# Motor Imagery Training Is Beneficial for Motor Memory of Upper and Lower Limb Tasks in Very Old Adults

**DOI:** 10.3390/ijerph20043541

**Published:** 2023-02-17

**Authors:** Pauline M. Hilt, Mathilde F. Bertrand, Léonard Féasson, Florent Lebon, France Mourey, Célia Ruffino, Vianney Rozand

**Affiliations:** 1INSERM UMR1093-CAPS, Université Bourgogne Franche-Comté, UFR des Sciences du Sport, F-21000 Dijon, France; 2Université Jean Monnet Saint-Etienne, Lyon 1, Université Savoie Mont-Blanc, Laboratoire Interuniversitaire de Biologie de la Motricité, F-42023 Saint-Etienne, France; 3Université Jean Monnet Saint-Etienne, CHU Saint-Etienne, Myology Unit, Referent Center for Neuromuscular Diseases, Laboratoire Interuniversitaire de Biologie de la Motricité, F-42023 Saint-Etienne, France; 4Institut Universitaire de France (IUF), F-75005 Paris, France; 5Laboratory Culture Sport Health and Society (C3S−UR 4660), Sport and Performance Department, University of Bourgogne Franche-Comté, F-25000 Besançon, France

**Keywords:** training, imagined contractions, aging, performance improvement, motor skills, memorization

## Abstract

Human aging is associated with a decline in the capacity to memorize recently acquired motor skills. Motor imagery training is a beneficial method to compensate for this deterioration in old adults. It is not yet known whether these beneficial effects are maintained in very old adults (>80 years), who are more affected by the degeneration processes. The aim of this study was to evaluate the effectiveness of a mental training session of motor imagery on the memorization of new motor skills acquired through physical practice in very old adults. Thus, 30 very old adults performed 3 actual trials of a manual dexterity task (session 1) or a sequential footstep task (session 2) as fast as they could before and after a 20 min motor imagery training (mental-training group) or watching a documentary for 20 min (control group). Performance was improved after three actual trials for both tasks and both groups. For the control group, performance decreased in the manual dexterity task after the 20 min break and remained stable in the sequential footstep task. For the mental-training group, performance was maintained in the manual dexterity task after the 20 min motor imagery training and increased in the sequential footstep task. These results extended the benefits of motor imagery training to the very old population, showing that even a short motor imagery training session improved their performance and favored the motor memory process. These results confirmed that motor imagery training is an effective method to complement traditional rehabilitation protocols.

## 1. Introduction

Aging is associated with progressive neural degenerative processes inducing impairments in cognitive function [1] such as declarative [2] or motor [3] memory. Motor memory, or memorization, is involved in motor learning through the encoding of planning details of the new-learned movement [4]. Although conflicting results exist in the literature on the age-related impairment of motor learning capacities [5,6], motor memory of new-learned movements is altered in the elderly after a 30 min break on a dexterity manual task [7] and after a 5 min break on a walking perturbation task [8]. Specifically, old adults showed a deterioration of performance after the resting period, whereas young adults preserved their performance compared to the performance measured before the break [7,8].

Although pharmacological interventions can reduce the age-related deterioration of motor memory [9], non-medical methods such as motor imagery training may be a relevant alternative to limit the use of drugs. Motor imagery is the mental simulation of a movement without overt execution [10]. Mental representation of an action shares similar neural networks with those activated during physical practice [11,12], like motor-related regions [13,14]. A large body of literature showed that motor imagery training over one or repeated sessions was effective in improving motor performance such as maximal strength [15,16] or motor skills [17,18]. In old adults, the ability to imagine simple movements is preserved but becomes more difficult for complex or unusual movements [19]. Hence, motor imagery training is still effective in improving motor performance in old adults [7,20]. Indeed, Ruffino et al. [7] showed that old adults who performed motor imagery training for 30 min preserved the gains obtained on a dexterity manual task, whereas those who rested for 30 min erased those gains. Whether these beneficial effects of motor imagery on motor memory are also maintained in very old adults (>80 years) remains unknown. Yet, very old adults are largely affected by the deterioration process of neuromuscular function [21] and would greatly benefit from this kind of rehabilitation technic. Although the temporal congruence between actual and imagined movements remains similar in young and old adults, imagined movement duration seems altered at very old ages (>80 years [22]). Similar results have been observed for old (62–67 years) and older individuals (71–75 years), with only the latter group showing significant differences in performance compared to young adults [23]. These results suggest that the beneficial effects of motor imagery on motor skill memorization may be lowered in very old adults.

Thus, the aim of the present study was to evaluate the effectiveness of a single session of mental training, using motor imagery, on the memorization of motor skill acquired by physical practice in very old adults. As in the study of Ruffino et al. [7], we tested this memorization process with a dexterity manual task (upper limbs). Due to the rapid deterioration of functional capacities in very old adults, notably in balance and mobility functions [24], we also tested the effects of mental training on a sequential footstep task (lower limbs). We hypothesized that motor imagery would have a positive effect on motor memory for both upper- and lower-limb tasks, but lower than that previously observed in old adults [7].

## 2. Methods

### 2.1. Participants

A total of 30 right-handed healthy very old adults (19 women, age: 86 ± 2 years, height: 163 ± 8 cm, weight: 62 ± 12 kg, Mini-Mental State Examination (MSSE) [25] = 29 ± 1, range [22,23,24,25,26,27,28,29,30]) volunteered to participate in the experiment and were clearly informed on the experimental procedures prior giving their written consent. Participants were recruited by phone call from the PROgnostic indicator OF cardiovascular and cerebrovascular events (PROOF) cohort, a prospective longitudinal cohort of old adults started in 2001. Participants had no neurological or physical disorders or disabilities and an MSSE score > 20. All the experimental procedures were approved by the French Ethics Committee Sud-Ouest et Outre-Mer 1 (ID-RCB: 2018-A02365-50) and were performed in accordance with the declaration of Helsinki, except for registration in a database (ClinicalTrials.gov, Identifier: NCT04018196). The University Hospital of Saint-Etienne was the sponsor of the present study.

### 2.2. Experimental Procedures

Participants, randomly assigned to a mental-training (*n* = 15) or a control (*n* = 15) group, visited the laboratory on two occasions separated by 24 h. During the first visit, participants completed an imagery ability questionnaire (mental-training group only) and functional tests (both groups, see below for details). Then, we assessed the effects of motor imagery training on a dexterity manual task. The participants were comfortably seated on a chair placed 20 cm in front of a table to perform a modified version of the nine-hole peg test (NHPT; Figure 1). The original NHPT was modified to increase the difficulty, duration and number of movements [7]. The NHPT required the participants to move 9 sticks as fast as possible into 9 holes in a pre-determined order and then replace them in a box. They started by moving the stick from hole 1 to hole A, then from hole 2 to hole B and so on. Once all sticks were placed into the corresponding holes, the sticks were moved into the box in the same order. The experimenter started the timer when the participant touched the first stick and stopped it when the last stick was in the box. Both groups performed 3 trials (a total of 108 movements) with 1 min rest between each trial during PreTest and PostTest. After performing the 3 PreTest trials, the participants either watched a non-emotional documentary for 20 min (*Home*, directed by Y. Arthus-Bertrand, 2009; control group) or performed mental training using motor imagery (mental-training group). The duration of the documentary was similar to the duration of the mental training. The participants of the mental-training group were instructed to imagine themselves performing the NHPT as fast as possible, combining the kinesthetic and visual (first-person perspective) modalities. They performed 3 blocks of 10 trials with 5 s rest between trials and 1 min rest between blocks to avoid mental fatigue, guided by the experimenter [26]. The experimenter tracked the duration of the rest periods (between trials and between blocks) using a stopwatch that was started as soon as the participants stopped imagining, and indicated to the participants when they should start the following imagined movement. All the participants performed the 30 imagined trials. The duration of the whole training protocol may have slightly varied between participants (±2 min) due to the variability in imagined movement duration and prior instructions. After pilot studies, the number of blocks was adjusted compared to a similar study of our research group on young and old adults [7] because the very old adults had difficulty staying focused during 5 blocks. Therefore, we reduced the number of blocks to 3 during the mental training. The duration of each imagined trial was measured to ensure isochrony between imagined and actual trials. The experimenter started the timer when the participant touched the first stick and stopped the timer when the participant dropped that same first stick.

During the second visit, performed at least 24 h after the first visit, we assessed the effects of mental training on skill performance improvement of lower limbs, using a sequential footstep task. To complete a sequence, the participants started in box 0, moved to box 1, moved back to box 0, then moved to box 2, moved up to box 4 and moved back to box 0. They were required to move their two feet in each box (surface of 0.25 m^2^) and performed two sequences per trial. The experimenter started the timer when the participant lifted the first foot and stopped it when the second foot touched the last box at the end of the trial. The participants performed 3 trials as fast as possible with 1 min rest between each trial at PreTest and PostTest. After performing the PreTest, the participants either watched the non-emotional documentary for 20 min (*Home*, directed by Y. Arthus-Bertrand, 2009; control group) or performed mental training using motor imagery (mental-training group). The participants of the mental-training group were instructed to imagine themselves performing the footstep task as fast as possible, combining the kinesthetic and visual (first-person perspective) modalities. They were sitting on a chair on box 0. They performed 3 blocks of 10 trials with 5 s rest between trials and 1 min rest between blocks (Figure 1). The duration of the imagined trials was assessed to ensure isochrony between imagined and actual trials. The experimenter started the timer when the participant lifted their right foot and stopped the timer when the participant put his foot down.

### 2.3. Imagery Ability

Participants of the mental-training group, but not the control group, completed a revised version of the Kinesthetic and Visual Imagery Questionnaire (KVIQ) [27] at the beginning of the first visit. It has to be noted that this test is part of the experimental intervention, as the two groups differed in this aspect. The KVIQ consisted of 10 simple movements involving either an upper limb, a lower limb or the trunk that were actually performed and then imagined with the visual and kinesthetic modalities. After each imagined movement from a first-person perspective, the participants rated on a 5-point scale the clarity of the images (visual mode) or the intensity of the sensations (kinesthetic mode) from 1 (no image/no sensation) to 5 (image as clear as when seeing the movement/sensation as intense as when executing the movement).

During mental training, the participants rated the vividness of their imagined movements on the same 5-point scale after each block.

### 2.4. Functional Tests

All participants performed the 6 min walk test (6MWT), which consisted of covering the longest distance possible in 6 min in a corridor of 30 m [28]. They also performed the timed up-and-go test (TUG), which required the participants to stand up from a chair, walk a distance of three meters, turn around, return and sit down on the chair [29]. The experimenter started the stopwatch when the back of the participant moved from the chair and stopped it when the participant sat with his/her back on the back of the chair.

### 2.5. Experiment on Old Adults

To compare the effect of mental training across ages, we used a dataset already published (Ruffino et al., 2019) using the NHPT task on a group of old adults (*n* = 13; mean age: 72 ± 4 years old, 7 females, MMSE = 29 ± 1). Ruffino and collaborators used the same task (modified version of the nine-hole peg test) in the same conditions. Their mental training was, however, longer (5 blocks of 10 trials). They also included a control group (*n* = 10; mean age: 74 ± 6 years old, 9 females, MMSE = 29 ± 1) who watched the same non-emotional documentary (*Home*, directed by Y. Arthus-Bertrand, 2009) for 30 min. Notably, old and very old participants had similar performances on the MMSE test (*p* = 0.79, t = 0.69).

### 2.6. Statistical Analysis

The Shapiro–Wilk test revealed that our dataset was not normally distributed (*p* < 0.05). For this reason, we used multiple two-tail permutation tests (5000 permutations) for all comparisons. The permutation test we used is based on a t-statistic. Depending on the algorithm chosen, it can be applied to paired or unpaired data. All analyses were performed using a custom software written in MATLAB (Mathworks). *p*-values were corrected for multiple comparisons using the Benjamini–Hochberg false discovery rate (MATLAB function fdr_bh).

Scores for the KVIQ and MMSE, TUG duration and 6MWT distance were compared between the mental-training and control groups, and movement durations were compared between imagined and actual trials in the mental-training group.

Initial performance between the two groups (mental-training vs. control) was compared within each PreTest trial and between the means of the 3 PreTest trials for NHPT and footstep task. Trial-by-trial evolution in the PreTest was evaluated in each group for both tasks.

To test the impact of mental training or the break period on performance retention, we compared the averaged movement duration between PreTest and PostTest, and also between the last PreTest trial (PreTest 3) and the first PostTest trial (PostTest 1) for the NHPT and footstep task. As an additional piece of information, we evaluated the slope of learning within PreTest and PostTest by computing linear regressions over the duration of the 3 trials. The slope values were then compared across groups to evaluate potential changes in learning rate after intervention.

Finally, to evaluate the difference of progression from PreTest to PostTest between our two groups of participants (mental-training vs. control), we computed for each participant and each task a normalized index of progression using the following formula:$$\frac{\mathrm{MPostTest}-\mathrm{MPreTest}}{\mathrm{MPreTest}}$$
where MPreTest and MPostTest designate the mean over the three PreTest and PostTest trials, respectively. This index was then compared across groups for each task.

Corrections were made within groups of comparison. For the evaluation of motor performance at PreTests, corrections were made on the six comparisons performed across groups (mental-training versus control for PreTest 1, PreTest 2, PreTest 3 and both tasks) and on the six comparisons performed within groups (PreTest 1 versus PreTest 2, PreTest 1 versus PreTest 3, PreTest 2 versus PreTest 3 and both tasks). For the evaluation of motor performance following the intervention, corrections were made on the eight comparisons performed between PreTest and PostTest (PreTest 1/PostTest 3 and MPreTest/MPostTest for both tasks and groups). For the learning slopes, corrections were made on the four comparisons performed between PreTest and PostTest (PreTest mental-training versus PreTest control and PostTest mental-training versus PostTest control for both tasks). For the progression index, corrections were made on the two comparisons performed between groups (NHPT mental-training versus NHPT control and Footstep mental-training versus Footstep control).

Data are presented as mean ± standard deviation in the text and table.

## 3. Results

### 3.1. Cognitive and Functional Capacities

Participants of the mental-training and control groups showed no significant difference in MMSE score (*p* = 0.37, t = 1.54), TUG duration (*p* = 0.84, t = 0.74) or 6MWT distance (*p* = 0.84, t = −0.82; Table 1).

The duration of the imagined movement (average across the 3 blocks; NHPT: 18.7 ± 9.4 s, footstep task: 16.7 ± 7.3 s) was not significantly different than the duration of the actual trials (average across PreTest and PostTest trials; NHPT: 18.2 ± 4.2 s, footstep task: 19.3 ± 5.3 s). Furthermore, the vividness of the imagined movements was close to the maximum score (4.67 ± 0.42), suggesting a good capacity to imagine.

### 3.2. Motor Performance at PreTests

For the modified version of the NHPT, permutation tests confirmed the absence of a significant difference in initial performance between the two groups (PreTest 1: *p* = 1.00, t = −0.11; PreTest2: *p* = 1.00, t = −0.17; PreTest3: *p* = 0.99, t = −0.37; mean PreTest: *p* = 0.84, t = −0.21). Within each group, the analyses revealed a longer duration in PreTest 1 (mental-training: 20.6 ± 3.9 s; control: 20.8 ± 4.3 s) compared to PreTest 2 (mental-training: 19.0 ± 4.1 s, *p* < 0.05, t = −3.65; control: 19.3 ± 3.5 s, *p* < 0.05, t = −2.54) and PreTest 3 (mental-training: 17.8 ± 4.1 s, *p* < 0.01, t = −4.10; control: 18.3 ± 2.5 s, *p* < 0.01, t = −3.44). PreTest 2 was significantly longer than PreTest 3 for the mental-training group only (*p* < 0.01, t = −4.10). The learning slopes computed over the three PreTest trials were similar between the two groups (mental-training slope: −1.4 ± 1.0; control slope: −1.2 ± 1.4; *p* = 0.99, t = −0.24).

For the footstep task, no significant difference was observed in initial performance between the two groups (PreTest 1: *p* = 0.51, t = 1.37; PreTest2: *p* = 0.48, t = 1.42; PreTest3: *p* = 0.82, t = 0.86; mean PreTest: *p* = 0.22, t = 1.31). Movement duration was longer in PreTest 1 (mental-training: 22.7 ± 5.3 s; control: 20.8 ± 2.9 s) compared to PreTest 2 (mental-training: 20.6 ± 5.4 s, *p* < 0.05, t = −3.10; control: 18.9 ± 2.8 s, *p* < 0.01, t = −4.49) and PreTest 3 (mental-training: 19.8 ± 5.5 s, *p* < 0.01, t = −3.83; control: 18.6 ± 3.3 s, *p* < 0.05, t = −2.81). No significant difference was found between PreTest 2 and PreTest 3 (mental-training: *p* = 0.22, t = −2.25; control: *p* = 0.98, t = −0.50). The learning slopes computed over the three PreTest trials were similar between the two groups (mental-training slope: −1.4 ± 1.5; control slope: −1.2 ± 1.4; *p* = 0.98, t = −0.41).

These results indicated that the two groups improved their performance through the PreTest trials in both upper- and lower-limb tasks.

### 3.3. Motor Performance Following the Intervention

Following the PreTest trials, the control group watched a documentary for 20 min to control for the evolution of the memorization process, while the experimental group performed a mental training consisting of 3 blocks of 10 trials (~20 min) to evaluate the effects of motor imagery-based training on the memorization of a new-learned skill. To obtain results comparable with the previous investigation on old people [7], we evaluated the progression for each group and each task between PreTest3 and PostTest1. To complement these results, we added a comparison between the mean of the PreTest trials and the mean of the PostTest trials. Finally, a standardized progression index was calculated to effectively compare the effect of our intervention between the two groups.

For the NHPT, the averaged movement duration across the three trials significantly decreased for the mental-training group (PreTest: 19.1 ± 3.9 s, PostTest: 17.2 ± 4.0 s, *p* < 0.001, t = −9.58), while no difference was observed for the control group (PreTest: 19.4 ± 3.2 s, PostTest: 18.5 ± 2.4 s, *p* = 0.48, t = −1.72). When performing a trial-by-trial analysis, movement duration was not significantly different from PreTest3 to PostTest 1 (*p* = 0.98, t = 0.72) for the mental-training group, but increased from PreTest3 to PostTest 1 (*p* < 0.05, t = 3.06) for the control group (Figure 2). We found no difference between the learning slopes computed over the three PostTest trials of the mental-training group (−0.8 ± 0.5) compared to the control group (−1.2 ± 0.9, *p* = 0.32, t = 1.70). Finally, when comparing the normalized index of progression across groups, we obtained a significant difference between the mental-training group and the control group (*p* < 0.05; t = −3.14). This result suggests that participants in the mental-training group (index of progression: −0.10 ± 0.05) improved their performance more in PostTest trials compared to PreTest trials than participants in the control group (−0.04 ± 0.09). Individual trajectories on the NHPT task are presented in Appendix A and highlight the consistency of the intervention effect. These results indicated that motor imagery training helped to prevent the motor memory deficit observed after a short break period for the control group.

For the footstep task, the averaged movement duration significantly decreased from PreTest trials to PostTest trials for the mental-training group (PreTest: 21.0 ± 5.2 s, PostTest: 17.5 ± 4.6 s, *p* < 0.001, t = −8.03), while no difference was observed for the control group (PreTest: 19.4 ± 2.7 s, PostTest: 17.9 ± 3.1 s, *p* = 0.09, t = −2.66). When performing a trial-by-trial analysis, movement duration significantly decreased from PreTest 3 to PostTest 1 (*p* < 0.01, t = −4.26) for the mental-training group, but was not significantly different from PreTest3 to PostTest 1 (*p* = 0.93, t = 0.91) for the control group (Figure 3). Linear regression over the duration of the three PostTest trials was lower for the mental-training group (−0.4 ± 0.6) than the control group (−1.2 ± 0.9, *p* < 0.05, t = 3.26). Finally, when comparing the normalized index of progression across groups, we obtained a significant difference between the mental-training group and the control group (*p* < 0.05; t = −3.49). This result suggests that participants in the mental-training group (index of progression: −0.16 ± 0.07) improved their performance more in PostTest trials compared to PreTest trials than participants in the control group (−0.08 ± 0.10). Individual trajectories on the footstep task are presented in Appendix A and highlight the consistency of the intervention effect.

For both tasks, no significant correlation was observed between the level of improvement and the score for KVIQ, MMSE, TUG or 6MWT (all *p* > 0.05).

### 3.4. Comparison with Old Adults

The data retrieved from Ruffino et al. (2019) showed that the averaged movement duration (NHPT) of old adults decreased from PreTest to PostTest trials only for the mental-training group (mental-training, PreTest: 16.6 ± 1.6 s, PostTest: 15.1 ± 2.0 s; control, PreTest: 19.4 ± 3.2 s, PostTest: 18.5 ± 2.4 s). This effect was marginally significant (mental-training: *p* = 0.053, t = −2.01), while the control group did not show any effect (*p* = 0.84, t = 0.20). This lack of statistical power after the mental training could be explained by the small number of participants included in Ruffino et al. (2019). However, the trend is consistent with the one we observed for very old adults.

Old adults showed shorter durations during the PreTest trials compared to very old adults (mental-training, old: 16.6 ± 1.6 s, very old: 19.1 ± 3.9 s, *p* < 0.05, t = −2.18; control, old: 13.6 ± 5.7 s, very old: 19.4 ± 3.2 s, *p* < 0.01, t = −3.38; Figure 4). After the break in the control groups (PostTest), very old adults were still slower than old adults (old: 14.1 ± 5.9 s, very old: 18.5 ± 2.4 s, *p* < 0.01, t = −2.64). After mental training, the movement duration was not significantly different between old and very old adults (old: 15.1 ± 2.0 s, very old: 17.2 ± 4.0 s, *p* = 0.09, t = −1.73). Furthermore, the performance of very old adults after mental training was similar to that of old adults after watching the non-emotional documentary (*p* = 0.12, t = −1.66).

## 4. Discussion

The aim of the present study was to evaluate the effectiveness of motor imagery training on motor memory in very old adults. Although performance was improved after three actual trials for both the dexterity manual task (upper limbs) and the sequential footstep task (lower limbs), the 20 min break period (control group) induced a decrease in performance for the dexterity manual task but no change for the sequential footstep task. However, very old adults who performed motor imagery training were able to keep their performance improvement for the dexterity manual task, and even increase their performance for the sequential footstep task.

### 4.1. Performance Improvement after Physical Practice

Initial performance on PreTest 1 was similar between the two groups of very old adults, ensuring a homogenous repartition of the participants between the control group and the mental-training group. Thus, we analyzed the improvement of motor performance throughout PreTest trials. The results showed that both the mental-training and control groups improved their performance across the three physical trials for the NHPT and footstep tasks, with a similar rate of improvement between groups. These results demonstrated that the ability to quickly improve motor performance is maintained at very old age, for both types of task. This finding confirmed previous observations showing that the ability to acquire novel motor skills was maintained with age [30,31]. In particular, it generalizes this observation to very old adults and full-body balance-related movements (footstep task). This is of interest, as the negative effects of aging on the neural structure involved in error-based motor learning have been previously described (e.g., cerebellum [32]). These neurophysiological declines may then affect the ability of seniors to improve their motor performance only in very particular tasks modalities. It has to be noted that the participants included in the present study were healthy and that their functional performance were better than the normative data for community-dwelling people aged over 80 years. Indeed, they walked farther during the 6 min walk test (men: ~455 m, women: ~420 m) than the normative data (men: 417 m, women: 392 m [33]), and the duration for the TUG test was similar or lower (men: ~8 s vs. 10 s; women: ~10 s vs. 11 s [33]). The sample of participants may not represent the whole population of octogenarians but only the healthiest and the most physically active ones.

### 4.2. Effects of the 20 Min Break Period on Performance

Despite performance improvement after three actual trials of the NHPT, very old adults returned to their initial performance after a 20 min break, erasing the improvement gains. This alteration in performance after a short break was already observed in old adults for the NHPT [7], thumb movements [9] or a locomotion task [8], whereas young adults were able to maintain their improvement gains. The deficit of motor memory in older adults was proposed to be due to alterations at the cerebral level. More precisely, the primary motor cortex (M1) is mainly involved in the memorization process [34,35], and some authors showed a dysfunction of this cortical area with aging [36,37]. For example, Jouvenceau et al. [36] observed in aged rats a decrease in NDMA receptors, which play a crucial role for motor memory and synaptic plasticity. Thus, this alteration at the cerebral level may partially explain the deficit of motor memorization in older adults. Interestingly, in the present study very old adults were able to maintain their performance at the sequential footstep task after the 20 min break. The better motor memory on the footstep task compared to the NHPT may not only be due to the limbs involved in the task (lower vs. upper limbs) because deficits in motor memory were already observed for the learning of a new walking pattern [8]. It would rather be due to the difficulty of the task, although we were not able to quantify or rate the difficulty of the two tasks. The NHPT requires fine motor skills to grab the sticks and place them in small holes repetitively, inducing a speed–accuracy tradeoff, whereas the footstep task involves moving the feet in large squares with sides 0.5 m in length, which requires dealing with balance but not precision. Further studies are needed to determine whether the limbs involved in the task or the characteristics of the task influence the memorization performance.

### 4.3. Effects of Mental Training on Performance

The main finding of the present study is the beneficial effect of motor imagery training on motor memory for both the NHPT and sequential footstep task in very old adults. Indeed, for the NHPT motor imagery training allowed the participants to avoid losing their performance improvement after a short break. For the footstep task, motor imagery training was even able to improve performance without actual practice. This result is in accordance with previous studies that showed a beneficial effect of motor imagery training on the lower-limb function of older adults. Indeed, a recent systematic review [38] provides evidence that motor imagery training improves balance and mobility in older adults. Importantly, the very old adults tested in the present study had a good capacity to imagine movements, which could have positively influenced the results [39], and the results could differ among participants having a lower capacity to form vivid movement images. These improvements in motor performance and motor memory may be directly linked with two components. Firstly, at a functional level, motor imagery practice seems to induce positive changes within motor planning [40,41]. Indeed, via the activation of the neural network implicated in motor planning, motor imagery training could help to refine the internal representation of the considered action by the generation of sensorimotor predictions, leading to an improvement of motor command [42,43,44]. Secondly, the modulation observed at both the neural [45] and spinal [46] levels during motor imagery seems to reinforce the sensibility and the conductivity of synapses in the corticospinal pathway [47], notably by increasing the effectiveness between pre- and post-synaptic neurons.

### 4.4. Efficiency of Mental Training across Ages

One of the strengths of the present study is the investigation of very old adults, for whom no data in motor imagery have been previously assessed. Such a terminological separation between old and very old adults may appear artificial, but greater alteration of neuromuscular function occurs after 75 or 80 years old [21,48]. In the present study, although very old adults show similar MMSE scores to those of old adults from the study of Ruffino et al. [7], they performed significantly slower on the NHPT task. Interestingly, more than being able to limit motor memory deficits, we showed here that mental training was able to compensate the loss of speed between very old and old adults. It has to be noted that to avoid fatigue, the length of the mental training was decreased for very old adults (3 blocks of 10 trials compared to 5 blocks of 10 trials for old adults [7]). Nonetheless, this shorter training duration was sufficient to improve performance. Taken together, these results suggest that a brief session of motor imagery training (~20 min) could be an efficient and low-cost rehabilitation method for very old adults, even with a lower number of repetitions compared to young and old adults. These results are encouraging for future experiments testing the effectiveness of a full training protocol over several weeks in very old adults.

## 5. Conclusions

The present study extended the benefits of motor imagery training to the very old population, showing that even a short motor imagery training session improves their performance and favors the motor memory process. These results confirmed that mental training, using motor imagery, is an interesting alternative option to complement traditional rehabilitation protocols. Mental training characteristics should be optimized in the future to maximize the beneficial effect of motor imagery in very old adults.

## Figures and Tables

**Figure 1 ijerph-20-03541-f001:**
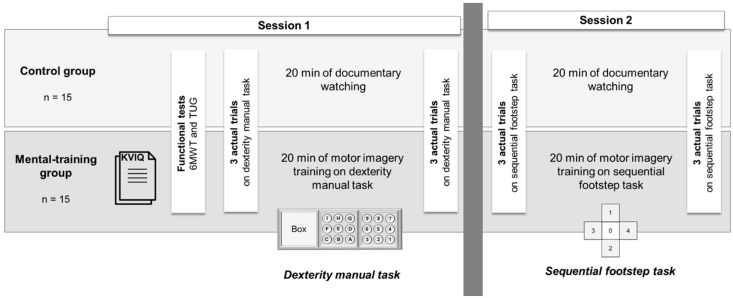
Illustration of the experimental protocol.

**Figure 2 ijerph-20-03541-f002:**
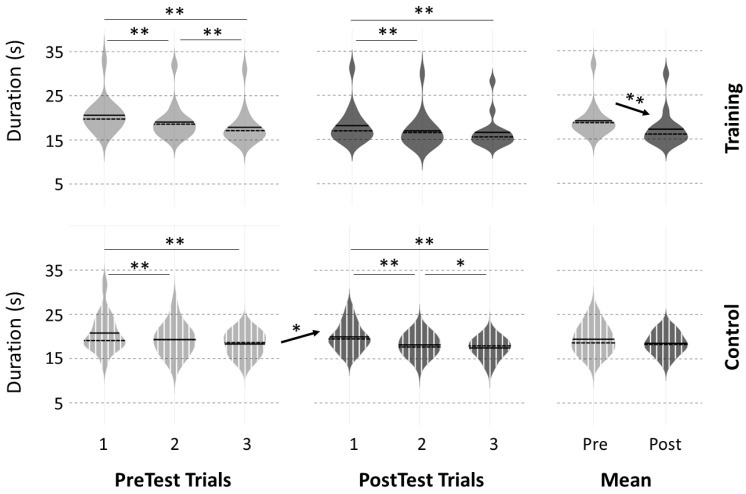
Violin representation of movement duration during the nine-hole peg test for each of the three PreTest and PostTest trials and the mean of the three trials for the mental-training and control groups. Significant difference: * *p* < 0.05, ** *p* < 0.01.

**Figure 3 ijerph-20-03541-f003:**
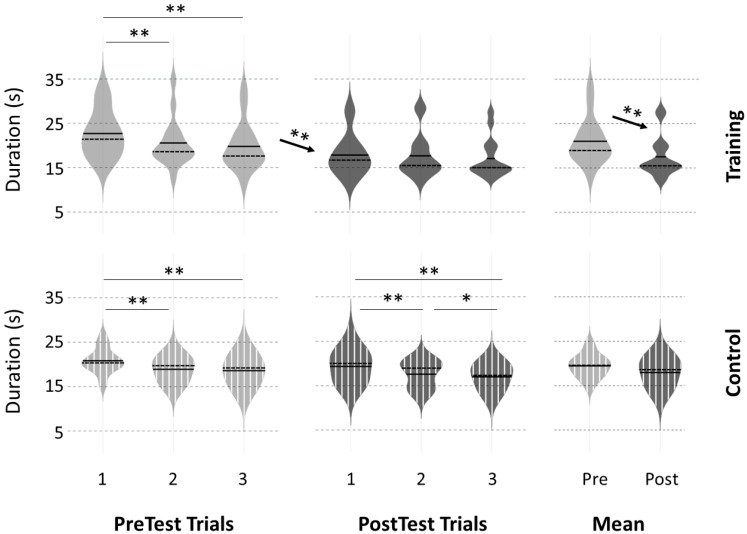
Violin representation of movement duration during the footstep task for each of the three PreTest and PostTest trials and the mean of the three trials for the mental-training and control groups. Significant difference: * *p* < 0.05, ** *p* < 0.01.

**Figure 4 ijerph-20-03541-f004:**
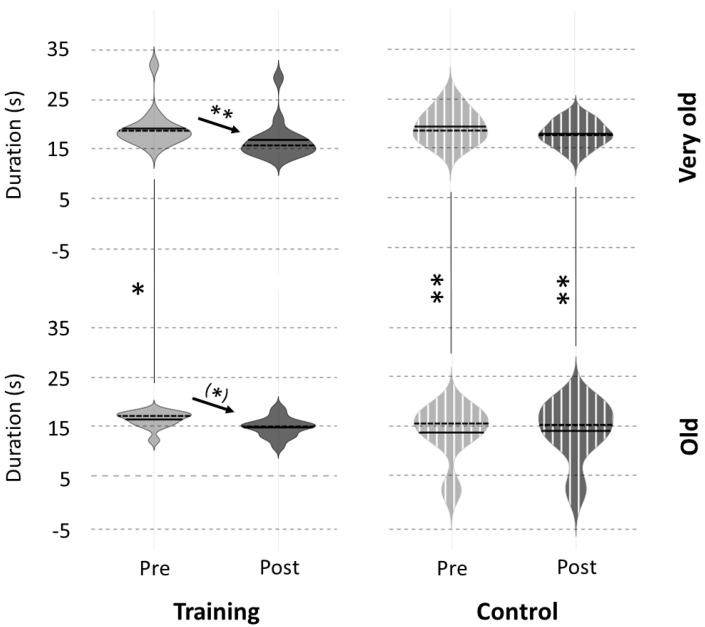
Comparison of movement duration on the NHPT task between our group of very old adults (age: 86 ± 2 years) and a group of old adults (age: 73 ± 5 years) extracted from Ruffino et al. (2019). Means of the three PreTest and PostTest trials are presented for each group. Significant difference: ^(^*^)^ *p* = 0.05, * *p* < 0.05, ** *p* < 0.01.

**Table 1 ijerph-20-03541-t001:** Cognitive and functional capacities of the mental-training and control groups. MMSE: Mini-Mental State Examination. KVIQ: Kinesthetic and Visual Imagery Questionnaire. 6MWT: 6 min walk test. TUG: Timed up-and-go test.

	Mental-Training Group	Control Group
MMSE	29 ± 1	29 ± 2
KVIQ	86 ± 15	
6MWT (m)	440 ± 68	426 ± 69
TUG (s)	9.0 ± 1.5	9.4 ± 2.0

## Data Availability

The data presented in this study are available on request from the corresponding author.

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
