# Peer review of "Motor Imagery Training Is Beneficial for Motor Memory of Upper and Lower Limb Tasks in Very Old Adults"

_ijerph, 2023, doi:10.3390/ijerph20043541_

Round 1
Reviewer 1 Report
The main research findings of this paper suggest that motor imagery training can be effective for exercise performance in older adults over the age of 80. This may be important for older adults who may have difficulty with physical exercise due to age-related declines in physical function and mobility. Motor imagery training is a non-physical form of exercise that involves mentally rehearsing movements, which may be a useful alternative for older adults who are unable to participate in traditional physical exercise.
Comment 1
It seems that the concern raised is that the subjects in the study may not be representative of the general older adult population, as they seem to have a very high level of physical function and cognition. To address this concern, the author of the paper would need to provide detailed information about how the subjects were recruited for the study. This should include information about the specific criteria used to select participants, as well as the recruitment process, including how participants were identified, contacted, and screened for inclusion in the study.
Comment 2
The present study compares data with Ruffino's 2019 data, but it is unclear whether direct comparisons of motor imagery performance were made in both studies; Ruffino's 2019 study used the MIQ-R and the present study used the KVIQ. Please provide a rationale for the use of different assessment.
Comment 3
It is unclear from the information in the paper whether the subjective estimates (SE) of motor imagery quality using the seven-point Likert scale used in Ruffino's study were also used to assess motor imagery vividness in the present study.
Did the present study assess the quality of motor imagery?
Comment 4
It is unclear from the information provided in the paper how the training was conducted and whether the protocol was controlled for all subjects equally.
To address this concern, the author should provide more information about the specifics of the 20 minutes of training in the MP training group. The author should mention whether the subjects took breaks individually or whether they were guided by the experimenter and, if so, how the breaks were managed. In addition, the author should mention whether the protocol was controlled equally for all subjects and whether there was any variation in the 20 minutes of training based on individual subjects.
It would also be important for the author to report whether there were any variations in the training protocol and whether these had any effect on the results of the study. In addition, the author should explain how the motor imagery training was standardised across subjects to ensure that the results are comparable.
Comment 5
I assume that mental fatigue was also taken into account, e.g. by including a one-minute break to eliminate the effects of fatigue caused by repetition of the exercise imagery, but what were the results in terms of actual fatigue levels? If you used VAS or other means to assess fatigue, please describe them.
Comment 6
In relation to age-specific motor imagery ability, please mention any studies that have investigated standard values of KVIQ for over 80s, etc., and how they compare with the results of such studies.
Comment 7
Line 80-. What was the basis for sample size of 30? Any power analysis?
Comment 8
Line175.I think that MIQ-R was mistake. It is not MIQ-R, but I think this sentence was KVIQ.
Author Response
Reviewer 1
The main research findings of this paper suggest that motor imagery training can be effective for exercise performance in older adults over the age of 80. This may be important for older adults who may have difficulty with physical exercise due to age-related declines in physical function and mobility. Motor imagery training is a non-physical form of exercise that involves mentally rehearsing movements, which may be a useful alternative for older adults who are unable to participate in traditional physical exercise.
Comment 1
It seems that the concern raised is that the subjects in the study may not be representative of the general older adult population, as they seem to have a very high level of physical function and cognition. To address this concern, the author of the paper would need to provide detailed information about how the subjects were recruited for the study. This should include information about the specific criteria used to select participants, as well as the recruitment process, including how participants were identified, contacted, and screened for inclusion in the study.
Response: Thank you for your valuable comment. The participants have been recruited from a local cohort of very old adults. We have specified the process of recruitment in the methods (L84-87):
“Participants were recruited by phone call from the PROgnostic indicator OF cardiovascu-lar and cerebrovascular events (PROOF) cohort, a prospective longitudinal cohort of old adults started in 2001. Participants had no neurological or physical disorders or disabilities and a MSSE score >20.”
We already mentioned in the discussion that the participants of the present study may not represent the whole population but rather the healthiest ones (L352-359):
“It has to be noted that the participants included in the present study were healthy and their functional performance were better than the normative data for community-dwelling people aged over 80 years. Indeed, they walked farther during the 6-min walk test (men: ~455 m, women: ~420m) than the normative data (men: 417 m, women: 392 m [33]) and the duration for the TUG test was similar or lower (men: ~8 s vs. 10 s; women: ~10 s vs. 11s [33]). The sample of participants may not represent the whole population of octogenarians but only the healthiest and the most physically active ones.”
Comment 2
The present study compares data with Ruffino's 2019 data, but it is unclear whether direct comparisons of motor imagery performance were made in both studies; Ruffino's 2019 study used the MIQ-R and the present study used the KVIQ. Please provide a rationale for the use of different assessment.
Response: Indeed, Ruffino’s 2019 study used the MIQ-R and we use the KVIQ. The MIQ-R has been developed and used to assess motor imagery ability in healthy adults and athletes, and its use in rehabilitation raises many difficulties. Because the high physical demands of several items, the MIQ-R is unsuitable for persons with physical disabilities (Malouin et al. 2007). The KVIQ has been developed for persons who, for different reasons, have to be guided in the rating of their imagery and who are not able to stand or to perform complex movements (Malouin et al. 2007). Even if our population above 80 years old did not have physical disabilities, we considered the KVIQ easier to perform and more adapted to our population.
Comment 3
It is unclear from the information in the paper whether the subjective estimates (SE) of motor imagery quality using the seven-point Likert scale used in Ruffino's study were also used to assess motor imagery vividness in the present study.
Did the present study assess the quality of motor imagery?
Response: We assessed the quality of imagery but not using the seven-point Likert scale. The completion of the KVIQ implies an assessment of the quality of motor imagery on a 5-point scale. In order to facilitate the comprehension and the use of the scale by the participants, we used the same 5-point scale to rate the vividness of motor imagery during the motor imagery training. This is specified in the methods (L158-159):
“During mental training, the participants rated the vividness of their imagined movements on the same 5-point scale after each block.”
The results of the vividness of motor imagery during the training are reported in the results (L231-232):
“Furthermore, the vividness of the imagined movements was close to maximum score (4.67 ±0.42), suggesting a good capacity to imagine.”
Comment 4
It is unclear from the information provided in the paper how the training was conducted and whether the protocol was controlled for all subjects equally.
To address this concern, the author should provide more information about the specifics of the 20 minutes of training in the MP training group. The author should mention whether the subjects took breaks individually or whether they were guided by the experimenter and, if so, how the breaks were managed. In addition, the author should mention whether the protocol was controlled equally for all subjects and whether there was any variation in the 20 minutes of training based on individual subjects.
It would also be important for the author to report whether there were any variations in the training protocol and whether these had any effect on the results of the study. In addition, the author should explain how the motor imagery training was standardised across subjects to ensure that the results are comparable.
Response: Thank you for your comment that will help clarifying the manuscript. The training protocol was standardized and guided by the experimenter. The participants were not able to take breaks when they wanted to, they had to follow the instruction of the experimenter. The experimenter controlled the duration of the rest periods (between trials and between blocks) with a stopwatch and indicated to the participants when they should start the following imagined movement. There can be slight variations (±2 min) in the duration of the whole training protocol due to the variability in the imagined movement durations between participants and the duration of the instruction prior to the training. However, this slight variation in the duration did not likely have an influence on the results as all the participants imagined the same number of movements. The manuscript now reads (L115-121):
“They performed 3 blocks of 10 trials with 5-s rest between trials and 1-min rest between blocks to avoid mental fatigue, guided by the experimenter [26]. The experimenter leaded the duration of the rest periods (between trials and between blocks) using a stopwatch that was started as soon as the participants stopped imagining, and indicated to the participants when they should start the following imagined movement. All the participants per-formed the 30 imagined trials. The duration of the whole training protocol may slightly vary between participants (±2 min) due to the variability in imagined movement duration and prior instructions.”
Comment 5
I assume that mental fatigue was also taken into account, e.g. by including a one-minute break to eliminate the effects of fatigue caused by repetition of the exercise imagery, but what were the results in terms of actual fatigue levels? If you used VAS or other means to assess fatigue, please describe them.
Response: Mental fatigue was not directly measured during the training protocol. We ensured the quality of motor imagery was good and not impacted by mental fatigue using the 5-point vividness score. This score did not change throughout the training protocol. We did not want to multiply the number of questionnaires or scores for simplicity for the very old participants.
Comment 6
In relation to age-specific motor imagery ability, please mention any studies that have investigated standard values of KVIQ for over 80s, etc., and how they compare with the results of such studies.
Response: Unfortunately there is no studies using the KVIQ with participants over 80 years old. To our knowledge, the only studied that tested participants over 80 in a specific group (Schott & Munzert, 2007) used another questionnaire: the Vividness of Movement Imagery Questionnaire. The other studies that used the KVIQ tested participants between 60 and 80 years old (e.g. Saimpont et al., 2015). Thus, we are not able to compare our results with the literature for the same age. However, it seems that the values of KVIQ reported by our participants is slightly higher than those reported by Saimpont et al. 2015 with participants aged ~70 years old. This difference may be due to the healthy and physically active profile of our participants or an overestimation of their capacity to clearly imagine the movements.
Comment 7
Line 80-. What was the basis for sample size of 30? Any power analysis?
Response: We performed an a priori analysis with GPower based on the results of Ruffino et al. 2019. This power analysis indicated that we should include 20 participants per group. However, we have had difficulties in the recruitment of very old adults, which is sometimes hard to make come to the lab, especially after the COVID-19 pandemic.
Comment 8
Line175.I think that MIQ-R was mistake. It is not MIQ-R, but I think this sentence was KVIQ.
Response: Thank you for the comment, we modified into the manuscript.

Reviewer 2 Report
I am sorry, but the main conclusions of the paper do not seem to be justified.
The authors interpret the absence of statistical significance as “maintaining performance”. This represents a well-known misuse of the concept of statistical significance.
Moreover, the authors interpret a significant change in the intervention group and lack of a significant change in the control group as evidence for a difference between the groups. This is simply incorrect. The correct way is to compare the mean change between the two groups (or equivalently to assess the interaction between time and group). A correct analysis will probably indicate no statistically significant differences, as for both tasks we observe a decrease in the outcomes from pre to post for both groups, with a rather small difference.
In addition, there are several points which should be addressed:
1. The participants had to volunteer for participation in the experiment. This may imply a biased selection of the participants. The authors mentioned explicitly a good capacity to imagine of the participant. How does this affect the generalizability of results?
2. The authors mentioned the use of a permutation test approach. This is adequate. However, permutation tests still depend on the test statistic used. These should be mentioned, in particular as both paired and unpaired data is analysed.
3. It is great to take multiplicity into account. However, for which collection of p-values was this done? Was there an attempt to correct for testing all hypotheses? Or does this only refer to the pair-wise tests (which are not very informative in this context).
4. I agree with the authors that the analysis of the data is somewhat tricky, as it is not straight forward how to take into account that there may be learning effects. The approach of the author to compare both overall mean values as well as the difference between pre 3 and post 1 makes some sense to me. (And I highly appreciate that the authors show a lot of raw data, such that it possible to get a rather broad picture.) However, it would be great if the authors could explain in the beginning how they intend to synthesize the results of the two test strategies.
5. It is a pity that the authors do not make more use of the fact that they have longitudinal data. I would highly recommend adding “spaghetti plots” allowing to follow the individual trajectories. It would make a difference whether all participants “maintain” their level of whether the picture is more diffuse.
6. The KVIQ was only applied in the mental-training group. This implies, that the baseline-assessment is part of the intervention tested. This should be clearly mentioned and may be a major limitation of the study.
7. The practical significance of the findings remains unclear. The experiment investigates a very-short term effect. How can such result inform the design of rehabilitation processes, which last many weeks and should improve long term outcomes? What is the practical recommendation from this experiment? Why does a comparison with pharmacological interventions make sense?
Author Response
Reviewer 2
I am sorry, but the main conclusions of the paper do not seem to be justified.
The authors interpret the absence of statistical significance as “maintaining performance”. This represents a well-known misuse of the concept of statistical significance.
Moreover, the authors interpret a significant change in the intervention group and lack of a significant change in the control group as evidence for a difference between the groups. This is simply incorrect. The correct way is to compare the mean change between the two groups (or equivalently to assess the interaction between time and group). A correct analysis will probably indicate no statistically significant differences, as for both tasks we observe a decrease in the outcomes from pre to post for both groups, with a rather small difference.
Response: Thank you for this remark, we chose this strategy of analysis to follow what was done by Ruffino et al 2019. However, we agree with the reviewer’s remark and added a between-group comparison to reinforce our results. To account for potential differences at the pre-test level, we decided to compare the progression within each group, normalized by the mean pre-test value for each subject. These elements were added in the method (L200-205) and results (L274-279 and L294-298).
In addition, there are several points which should be addressed:
- The participants had to volunteer for participation in the experiment. This may imply a biased selection of the participants. The authors mentioned explicitly a good capacity to imagine of the participant. How does this affect the generalizability of results?
Response: We added some information about the selection process in the methods L84-86 and we discussed the possible selection bias in the discussion L352-359.
We also added a sentence in the discussion to indicate that the results may differ with participants having a lower capacity to form vivid movement images (L392-394):
“Importantly, the very old adults tested in the present study had a good capacity to imagine movements, which could positively influence the results [39], and the results could differ with participants having a lower capacity to form vivid movement images.”
- The authors mentioned the use of a permutation test approach. This is adequate. However, permutation tests still depend on the test statistic used. These should be mentioned, in particular as both paired and unpaired data is analysed.
Response: Thank you, we added this information into the manuscript (L181-183).
“The permutation test we used is based on a t-statistic. Depending on the algorithm chosen, it can be applied to paired and unpaired data.”
- It is great to take multiplicity into account. However, for which collection of p-values was this done? Was there an attempt to correct for testing all hypotheses? Or does this only refer to the pair-wise tests (which are not very informative in this context).
Response: The corrections were made within groups of comparison. We added this information in the manuscript (L206-218)
“The corrections were made within groups of comparison. For the evaluation of motor performance at PreTests, the corrections were made on the six comparisons performed across groups (mental-training versus control for PreTest 1, PreTest 2, PreTest 3 and both tasks) and on the six comparisons performed within groups (PreTest 1 versus PreTest 2, PreTest 1 versus PreTest 3, PreTest 2 versus PreTest 3 and both tasks). For the evaluation of motor performance following the intervention, the corrections were made on the eight comparisons performed between PreTest and PostTest (PreTest 1/PostTest 3, MPre-Test/MPostTest for both tasks and groups). For the learning slopes, the corrections were made on the four comparisons performed between PreTest and PostTest (PreTest mental-training versus PreTest control, PostTest mental-training versus PostTest control for both tasks). For the progression index, the corrections were made on the two comparisons performed between groups (NHPT mental-training versus NHPT control and Footstep mental-training versus Footstep control).”
- I agree with the authors that the analysis of the data is somewhat tricky, as it is not straight forward how to take into account that there may be learning effects. The approach of the author to compare both overall mean values as well as the difference between pre 3 and post 1 makes some sense to me. (And I highly appreciate that the authors show a lot of raw data, such that it possible to get a rather broad picture.) However, it would be great if the authors could explain in the beginning how they intend to synthesize the results of the two test strategies.
Response: Thank you for this remark. To clarify this point, we have added a few lines describing our objectives in the results (L260-265).
“To obtain results comparable with the previous investigation on old people [7], we evaluated the progression for each group and each task between PreTest3 and PostTest1. To complement these results, we added a comparison between the mean of the PreTest trials and the mean of the PostTest trials. Finally, a standardized progression index was calculated to effectively compare the effect of our intervention between the two groups.”
- It is a pity that the authors do not make more use of the fact that they have longitudinal data. I would highly recommend adding “spaghetti plots” allowing to follow the individual trajectories. It would make a difference whether all participants “maintain” their level of whether the picture is more diffuse.
Response: We thank the reviewer for this remark. Below are “spaghetti plots” for the two tasks and two groups. We think the violin representation for these results is easier to understand the changes in the means, and we could add the figures below in a supplementary material file, but we would accept to add these figures in the text if the reviewer thinks it is necessary.
Figure S1: Individual trajectories across the 3 PreTest and PostTest trials for the mental-training (A) and control (B) groups, in the NHPT task.
Figure S2: Individual trajectories across the 3 PreTest and PostTest trials for the mental-training (A) and control (B) groups, in the Footstep task.
- The KVIQ was only applied in the mental-training group. This implies, that the baseline-assessment is part of the intervention tested. This should be clearly mentioned and may be a major limitation of the study.
Response: The KVIQ was performed at the beginning of the session, before the functional tests, and the movements that are evaluated are completely different from those performed during the intervention. Furthermore, the duration of the PreTest trials was similar between the two groups, ensuring that the completion of the KVIQ did not influence the performance at the NHPT. These arguments make us confident that this is not a major limitation of the study. This setup is identical from the one used in Ruffino et al. (2019), with only the mental-training group completing the questionnaire, which allowed us to make the direct comparison between the participants of the two studies.
We now clearly specified in the Methods that only the mental-training group completed the KVIQ (L149):
“Participants of the mental-training group, but not the control group, completed the revised version of the Kinesthetic and Visual Imagery Questionnaire (KVIQ)”
- The practical significance of the findings remains unclear. The experiment investigates a very-short term effect. How can such result inform the design of rehabilitation processes, which last many weeks and should improve long term outcomes? What is the practical recommendation from this experiment? Why does a comparison with pharmacological interventions make sense?
Response: The present study is the first step to demonstrate the effectiveness of motor imagery training in very old adults. There are two valuable information from this experiment that can help the design of a prevention of rehabilitation program using motor imagery: 1/ motor imagery is still efficient at very old age and can be used as a rehabilitation technic in this population, and 2/ the duration of the training sessions should be shortened compared to young and old adults, with still beneficial effects. We agree with the reviewer that this single-session training can not be generalized to a long term protocol. We have now indicated at the end of the discussion that this kind of protocol would need to be tested over several weeks (L415-420):
“Taken together, these results suggest that a brief session of motor imagery (~20 min) could be an efficient and low cost rehabilitation method for very old adults, even with a lower number of repetitions compared to young and old adults. These results are encouraging for future experiments testing the effectiveness of a full training protocol over several weeks in very old adults.”
The comparison with pharmacological interventions in the Introduction was presented for the general method of motor imagery over a full training protocol. Motor imagery (in general) could lower the use of drugs for prevention or rehabilitation, which is important considering the deleterious or inefficient effects of drug cocktails, common at very old age. Thus, this experiment is the first step to develop this kind of non-medical method in the future.

Round 2
Reviewer 1 Report
Thank you for your prompt and comprehensive response to the peer review comments.
I am pleased to accept the manuscript.
Author Response
We thank the reviewer for his previous comments that helped improving the manuscript.
Reviewer 2 Report
Thanks a lot for your careful revision. I was very surpised to see that there is a statistically significant difference between the two groups when making the essential comparison. But this is of course great.
The standard deviations you reported for the normalized index are rather low in the intervention group. This explains on the one side the statistical significance. On the other side, it also tells us that the intervention effects are very uniform, i.e. the vast majority of patients experience an improvement. ( I suppose this is also visible in the spaghetti plots. I have unfortunately no access to the supplement. (And it is of course fine to present these graphs in the suppelement.)) Perhaps the uniformity of the intervention effect is an aspect the authors would like to mention. This seems to me a further, important property of the intervention.
With respect to the KVIQ, I do neither regard this as a major limitation. The point is more that it would be fair to describe this is a part of the experiemntal intervention, as the two groups differ in this aspect. This does not seem to be a problem, as it is rather natural to start with this assessment.
Author Response
Thanks a lot for your careful revision. I was very surpised to see that there is a statistically significant difference between the two groups when making the essential comparison. But this is of course great.
Response: We thank the reviewer for his further comments that helped us improving our manuscript. We addressed below the two questions
The standard deviations you reported for the normalized index are rather low in the intervention group. This explains on the one side the statistical significance. On the other side, it also tells us that the intervention effects are very uniform, i.e. the vast majority of patients experience an improvement. ( I suppose this is also visible in the spaghetti plots. I have unfortunately no access to the supplement. (And it is of course fine to present these graphs in the suppelement.)) Perhaps the uniformity of the intervention effect is an aspect the authors would like to mention. This seems to me a further, important property of the intervention.
Response: Thank you for this comment, we now mentioned and discussed these results in the results part (L280-281 and L300-301). We also added these two figures in supplementary materials.
With respect to the KVIQ, I do neither regard this as a major limitation. The point is more that it would be fair to describe this is a part of the experiemntal intervention, as the two groups differ in this aspect. This does not seem to be a problem, as it is rather natural to start with this assessment.
Response: Thank you for this comment, we now clarified this point in the method part (L151-152).